# Mixed Precision Training With 8-bit Floating Point

## Abstract

Reduced precision computation is one of the key areas addressing the widening 'compute gap', driven by an exponential growth in deep learning applications. In recent years, deep neural network training has largely migrated to 16-bit precision, with significant gains in performance and energy efficiency. However, attempts to train DNNs at 8-bit precision have met with significant challenges, because of the higher precision and dynamic range requirements of back-propagation. In this paper, we propose a method to train deep neural networks using 8-bit floating point representation for weights, activations, errors, and gradients. We demonstrate state-of-the-art accuracy across multiple data sets (imagenet-1K, WMT16) and a broader set of workloads (Resnet-18/34/50, GNMT, and Transformer) than previously reported. We propose an enhanced loss scaling method to augment the reduced subnormal range of 8-bit floating point, to improve error propagation. We also examine the impact of quantization noise on generalization, and propose a stochastic rounding technique to address gradient noise. As a result of applying all these techniques, our models generalize as well as the full precision models, and achieve baseline accuracy.

## 1 Introduction

The unprecedented success of Deep Learning models in a variety of tasks including computer vision (He et al., 2016), machine translation (Wu et al., 2016) and speech recognition (Graves et al., 2013; Hannun et al., 2014) has led to the proliferation of deeper and more complex models. Algorithmic innovations such as large batch training (Keskar et al., 2016) and neural architecture search (Zoph & Le, 2016) have enabled models to scale on large compute cluster to accelerate training. This enhanced performance has enabled the adoption of larger neural networks. As a consequence, the computational requirements for training Deep Learning models have been growing at an exponential rate (Amodei & Hernandez) over the past few years, outperforming Moore's Law and hardware capabilities by a wide margin.

One of the promising areas of research to address this growing compute gap is to reduce the numeric precision requirements for deep learning. Reduced precision methods exploit the inherent noise resilient properties of deep neural networks to improve compute efficiency, while minimizing the loss of model accuracy. Recent studies (Micikevicius et al., 2017; Das et al., 2018) have shown that, deep neural networks can be trained using 16-bits of precision without any noticeable impact on validation accuracy across a wide range of networks. Today, state-of-the-art training platforms support 16-bit precision in the form of high-performance systolic array or GEMM engine (General Matrix Multiply) implementations (Markidis et al., 2018; Köster et al., 2017a).

There have been numerous attempts (Hubara et al., 2017; Zhou et al., 2016; De Sa et al., 2018; Wu et al., 2018; Cai et al., 2017) to train deep neural networks at lower precision (below 16-bits) with varying degrees of success. With the abundance of 8-bit integer deep learning 'ops' deployed to accelerate inference tasks, much of the research into training methods have also focused on integer based fixed-point numeric formats (Zhou et al., 2016; De Sa et al., 2018; Wu et al., 2018). Training with 8-bit integers has been significantly more challenging because the dynamic range of such formats is not sufficient to represent error gradients during back-propagation. More recently, Wang et al. (2018) have shown that 8-bit floating representation can be used to train convolutional neural networks, with the help of specialized chunk-based accumulation and stochastic rounding hardware.

While this method has shown promising results, it requires expensive stochastic rounding hardware built into the critical compute path making it unattractive for systolic array and GEMM accelerator implementations.

Our paper extends the state of the art in 8-bit floating point (FP8) training with the following key contributions:

- We propose a scalable training solution that eliminates the need for specialized hardware designs (Wang et al., 2018), thereby enabling efficient MAC designs with higher compute density.

- We demonstrated state-of-the-art training results using 8-bit floating point representation (for weight, activation, error and gradient tensors), across multiple data sets (Imagenet-1K, WMT16) and a broader set of workloads (Resnet, GNMT, Transformer) than previously reported (Wang et al., 2018).

- We propose enhanced loss scaling method to compensate for the reduced subnormal range of 8-bit floating point representation for improved error propagation leading to better model accuracy.

- We present a detailed study of the impact of quantization noise on model generalization and propose a stochastic rounding technique to address the gradient noise in the early epochs leading to better generalization.

## 2 RELATED WORK

The study of reduced precision methods for deep learning training is an active area of research. In the pursuit of improving compute efficiency, researchers have experimented with various numeric formats and hardware implementations. Gupta et al. (2015) demonstrated that deep neural networks can be trained with minimal loss in accuracy, using 16-bit fixed point representation. This was followed by studies employing other numeric formats such as, half-precision floating point (Micikevicius et al., 2017) and dynamic fixed point (Köster et al., 2017b; Das et al., 2018), demonstrating state of the art results across residual (He et al., 2016), recurrent (Wu et al., 2016) and generative networks. Today most of the neural network training in a production deployment has migrated to 16-bit hardware, resulting in significant improvements in performance (Markidis et al., 2018).

There have been several attempts to further reduce the precision requirements of DNNs to boost training performance. Zhou et al. (2016) have trained DoReFa-Net, a derivative of AlexNet (Krizhevsky et al., 2012) using bit-convolutions with 1-bit and 2-bits to represent weights and activations respectively, while the gradients are quantized to 6-bits of precision. Wu et al. (2018) have trained AlexNet (Krizhevsky et al., 2012) using 8-bit precision for activations, errors and weight gradients, while the weights are quantized to 2-bits of precision. However, both these methods have reported significant loss in validation accuracy.

More recently, Wang et al. (2018) have successfully trained Resnet-50 (He et al., 2016) using 8-bit floating point numeric format with the help of a specialized hardware to compute chunk-based dot-product computation and stochastic rounding on a 16-bit accumulator. The authors of this study have focused on reducing the accumulator precision and based on studies on smaller networks (AlexNet Resnet-18), attributed training issues related to error propagation and generalization on the choice of accumulator size. However, our studies on larger networks (Resnet-34/50) using 32-bit accumulator indicate that, these issues are not related to the choice of accumulator size and should be addressed independently. We discuss these issues and our proposed solutions in greater detail in Sections3.1and 3.2. Guided by these results, we decided to focus on studying the impact of using FP8 numeric format on training, while maintaining a high precision accumulator(FP32). We further believe that modern GEMM engine designs implementing progressive multiplier reduction (Ibrahim & Gebali, 2016) techniques can effectively amortize the cost of a larger final accumulator, and do not benefit from building 16-bit accumulator (Wang et al., 2018) with additional hardware overheads in the critical compute path.

## 3   TRAINING METHOD

The choice of bit-level representation of floating point (**sign, exponent, mantissa**), has a significant impact on the effectiveness of the numerical format – the trade-off between the dynamic range and precision is especially tricky at low bit-width representations. While it is important to maintain higher dynamic range for effective propagation of error gradients(Micikevicius et al., 2017), it leads to having values that are too few and scattered to maintain fidelity required for gradient computations. After careful consideration of these facts and several failed experiments with other formats (ex: with more exponent bits), we decided to use **s=1,e=5,m=2** numeric format for representing 8-bit floating point. We also use a 32-bit floating point accumulator to accommodate the dynamic range of the output resulting from the dot-product operation. Therefore, each GEMM/convolution operation takes two input tensors in 8-bit floating point format and produces a 32-bit single precision floating point output. The 32-bit output must be down-converted to a 8-bit floating point value before passing to the next operation. We believe rounding plays an extremely important role during down-conversion operation to help recover the numeric accuracy. We present the results from the study of different rounding modes applied to this numeric format and their impact on training performacne in Section.3.2

Figure.1a shows the precision settings of various compute operations used in our mixed precision training setup. The **'GEMM'** operator shown in Figure.1a represents the key compute kernel used by deep neural networks during forward, backward, and gradient computation passes. Quantization nodes identified with the letter **'Q'** perform down-conversion and rounding on 32-bit output from GEMM operators and convert them them to 8-bit floating point format before passing on to the next layer. In our training experiments, we quantized the weight, activation, error and gradient tensors of all convolution and MatMul kernels to 8-bit floating point format in forward, backward and weight update paths. More details about workload specific quantization flow is discussed in Section 4.

Figure.1b shows the data flow during optimization and weight update steps. In the optimization path the L2-regularization term is added to the cross entropy, and the resulting loss is multiplied with *loss_scale* parameter before initiating back propagation. The weight gradients are computed during back propagation and converted to 8-bit floating point format. During weight update, the weight gradients are re-scaled using the *loss_scale* parameter, this step is performed in full precision to prevent any potential underflow. The gradients are then passed to the momentum optimizer, and the final gradients are applied to the master copy of weights. In our experiments, the master copy of weights are stored in half-precision floating point format, these values are up-converted to 32-bit during the update step and the weight update is performed at full precision. After the update, the master weights are converted to half-precision format before they are stored back into memory. Since this is a bandwidth bound operation, compute precision will not have any noticeable impact on the performance.

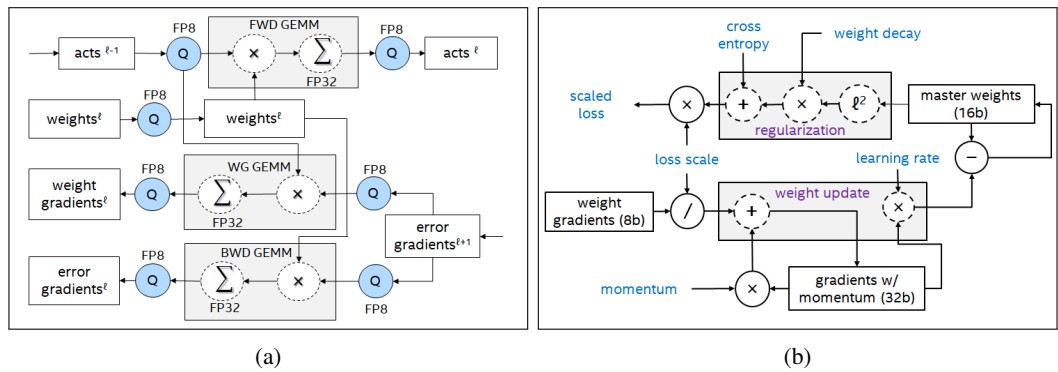

(a)                                    (b)

Figure 1: Mixed precision data flow for FP8 training. (left) precision settings for key compute kernels in Forward, Backward, and Weight Update passes, (right) flow diagram of the weight update rule.

## 3.1 Enhanced Loss Scaling

Previous studies (Micikevicius et al., 2017) on half-precision floating point have shown that loss scaling technique can be used to push smaller error gradients into representable range and train neural networks successfully. The full range of numeric values represented by a floating point format include the 'subnormal' values, the range of which is determined by the number of mantissa bits. Because of this property of floating point numbers, the proposed 8-bit floating point format will have significantly smaller subnormal range compared to a half-precision floating point which has the same number of exponent bits. Table.1 shows the dynamic range comparison between full-precision(FP32), half-precision(FP16) and the proposed 8-bit floating point (FP8) formats.

Table 1: Dynamic range comparison between proposed FP8 and other floating point formats.

| Data Type | Bit Format (s, e, m) | Max Normal | Min Normal | Min Subnormal |
|---|---|---|---|---|
| IEEE-754 float | 1, 8, 23 | 3.40e38 | 1.17e−38 | 1.40e−45 |
| IEEE-754 half-float | 1, 5, 10 | 65 535 | 6.10e−5 | 5.96e−8 |
| FP8 (proposed) | 1, 5, 2 | 57 344 | 6.10e−5 | 1.52e−5 |

Half-precision training for convolution networks has been shown to converge using a constant loss scaling value (ex: Resnet-50 uses 1000)(Micikevicius et al., 2017). Networks like GNMT (Wu et al., 2016) and Transformer (Vaswani et al., 2017) experience significant variations in gradient distributions through the training cycle and perform better with dynamic loss scaling methods. We adopt the same loss scaling methods used by FP16 training for FP8. For Resnet-50, we increase the scaling factor to compensate for the smaller subnormal range of FP8. Figure.2a shows results from our convergence studies on Resnet-50 using different loss scaling values. The model failed to converge with a scaling factor of 1000, and progressively performed better with increasing loss scale values, converging at 10 000. Transformer (big) model, uses the standard 'DynamicLossScale' method implemented in TensorFlow and it works **out-of-the-box** without any additional changes. We use the OpenSeq2Seq (Kuchaiev et al., 2018) implementation for GNMT model, and it uses 'back-off' method for automatic loss scaling. Both these loss scaling methods try to maintain a higher loss scale value while checking for numeric overflows at regular intervals(=2000), that result in a 'NaN' during gradient computation. Our experiments showed that loss scaling experiences frequent gradient overflows for GNMT because of a few outliers, while a significant chunk of the gradients underflow. This irregular gradient distribution is likely a result of not having normalization layers in GNMT. The loss scaling algorithm responds to these overflows by dropping the loss scale value significantly. We addressed this by ignoring a few overflows which are likely a result of the outliers and continue to maintain a higher loss scale value. We accomplish this by setting a 'minimum threshold' for the loss scale value to prevent it from going below a certain threshold value even if an overflow occurs. Figure.2b shows the loss scaling schedule that worked for GNMT.

The goal of this exercises was to maintain a higher loss scale value, while ignoring a few spurious overflows. This can also achieved by enhancing the loss scaling algorithm with few additional checks to ignore overflows unless they occur in succession for a few times that is predefined threshold value (nan_threshold). Overflows occurring at successive intervals is a more reliable indicator of a actual shift in the gradient distribution. We can also reduce the interval between loss scale updates (from 2000 iterations to 500 iterations), to improve chances of recovering from any inadvertent drop in loss scale value. We describe this algorithm in 1.

## 3.2 Quantization noise and Generalization

Reduced precision methods introduce significant amount of noise that can adversely effect convergence and accuracy of deep neural networks. Rounding techniques applied to quantization methods can be effective in regulating some of this noise. For extremely low precision representations with large rounding errors such as the one proposed here($\epsilon = 0.125$), the choice of rounding method can have significant influence on the numeric accuracy and overall applicability of the numeric format. Previous studies (Gupta et al., 2015) have shown that stochastic rounding can be effective for training neural networks using low-precision fixed point formats. The most widely supported rounding

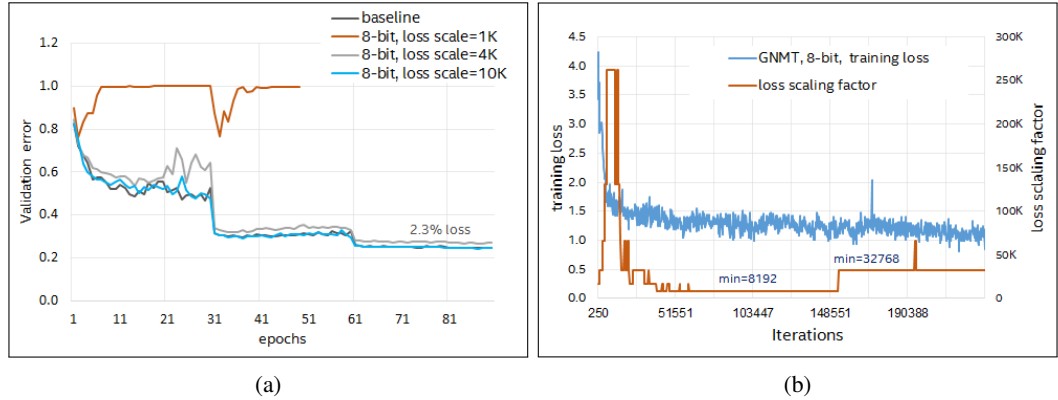

(a)                          (b)

Figure 2: Convergence behaviour of FP8 training using enhanced loss scaling. (left) Resnet-50 (He et al., 2016) failed to converge with loss scale=1000, performed better with 2.3% accuracy loss at loss scale=4000 and showed full convergence at loss scale=10 000, (right) Dynamic loss scaling with gradually increasing minimum threshold for the scaling factor.

---

**Algorithm 1** Enhanced loss scaling algorithm

---

**Ensure:** $minval = 2, maxval = 2^{15}, interval = 500, scale = 2, nan\_threshold = 2$
 1:  $nan\_count = 0, last\_iter = 0, last\_nan\_iter = 0$
 2:  **procedure** UPDATE_LOSS_SCALE (loss_scale, has_nan, iter)
 3:     **if** $has\_nan = True$ **then**
 4:         **if** $nan\_count >= nan\_threshold$ **then**
 5:             $loss\_scale \leftarrow loss\_scale \div 2$
 6:             $nan\_count \leftarrow 0$
 7:         **else if** $iter = last\_nan\_iter + 1$ **then**
 8:             $nan\_count \leftarrow nan\_count + 1$
 9:     **else if** $iter - last\_iter >= interval$  **then**
10:         $loss\_scale \leftarrow loss\_scale \times 2$
11:     **return** loss_scale

---

method in hardware today is RNE (round to nearest even), because it is easier to implement and requires smaller silicon area. In this section, we explore the impact of both RNE and stochastic rounding methods on model convergence and generalization.

Our early experiments showed that, for smaller networks such as Resnet-18 (He et al., 2016), RNE proved quite effective when trained on Imagenet-1K (Deng et al., 2009) data set. However, when we trained ResNet-50 (He et al., 2016) we observed some interesting results. Figure.3 shows the convergence plots for Resnet-50 (He et al., 2016) using RNE rounding method applied to quantized weights, activations and gradients. The model displayed significant over-fitting behaviour as indicated by the increased validation error, while the training error mostly follows the baseline as shown in as shown in Figure.3b, and 3a. Multiple experiments indicated that this behaviour is caused by the noisy error gradients during early epochs which lead to unconstrained growth in model parameters. This is indicated by steep increase in L2 regularization parameter as shown in Figure.3c. Regularization loss is computed using the formula shown in Equation.1. Increased regularization loss leads to more noisy gradients, which further exacerbates this behaviour.

An interesting observation about the L2 regularization loss is that for ResNet-18, the L2-loss is low at the beginning and increases with gradually with iterations. On the other hand for ResNet-50, the L2-loss is high at the beginning due to the initialization of low fan-in 1x1 (Glorot & Bengio, 2010) convolutions, and needs to dip a little before gradually rising again. We suspect that this property of

the initialization leads to more noisy behavior of ResNet-50 in the earlier iterations as compared to ResNet-18. Therefore for the ResNet-50 model stochastic rounding is essential.

$$L2\_loss = \lambda \times \sum_{i=0}^{W} w_i^2 \tag{1}$$

Where, $\lambda$ is the weight decay parameter and W is the total number of weights.

In order to understand the issue of regularization independent of the choice of rounding method, we conducted additional experiments using RNE with other forms of regularization. Figure.4a compares the 'Dropout' method with 'no regularization' method which uses quantization noise as implicit regularizer with no explicit regularization term. In both these cases, the models performed much better than using L2 regularization with RNE, leading us to the conclusion that RNE was ineffective in regulating quantization noise in gradients causing unconstrained growth in model parameters.

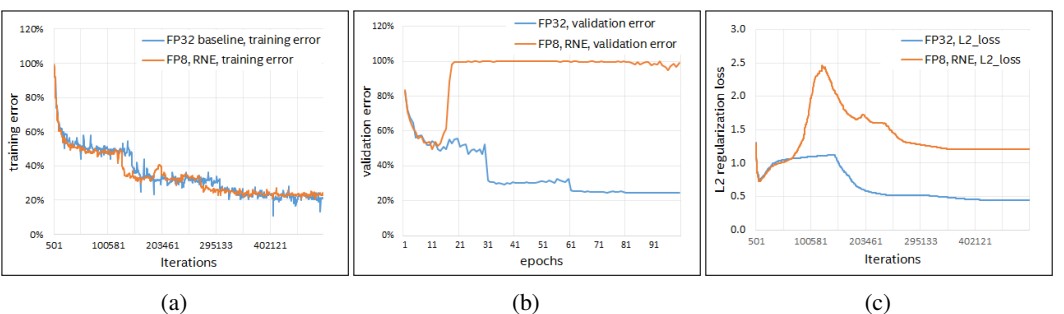

(a)                              (b)                              (c)

Figure 3: Impact of quantization (with RNE rounding) noise on model convergence with Resnet-50 (a) comparison of training error, (b) validation error, and (c) L2 regularization loss with FP32 baseline.

Unlike deterministic rounding techniques, stochastic rounding computes the probability of rounding using information from several discarded bits of the input making it less prone to introducing large rounding errors. We studied the error behaviour of Resnet-50 (He et al., 2016) by applying stochastic rounding on activations and gradients to regulate quantization noise in the gradients, which in-turn can improve the effectiveness of explicit regularization methods.

Our stochastic rounding method is defined as follows:

$$round(x, k) = \begin{cases} \lfloor x \rfloor_k + \epsilon, & \text{with probability } P = \frac{(x - \lfloor x \rfloor_k) + r}{\epsilon} \\ \lfloor x \rfloor_k, & \text{with probability } 1 - P \end{cases}$$

Where, k is the target precision, $\epsilon$ is machine epsilon, and $r$ is random value generated by a pseudo random number generator.

Figure.4b shows the results from Resnet-50 (He et al., 2016) training experiment using a combination stochastic rounding and explicit L2 regularization. The convergence plots show the good generalization behavior that tracks with the full precision training. The accuracy numbers are summarized in Section.4.

## 4    EXPERIMENTS AND RESULTS

We built a TensorFlow based training platform that can accurately emulate the numeric properties of 8-bit floating point on existing floating point hardware. Training experiments were conducted using open source model implementations from TensorFlow[1] and OpenSeq2Seq (Kuchaiev et al., 2018). Our training framework internally updates the training graph by inserting quantization nodes

---

[1]Models: https://github.com/tensorflow/models

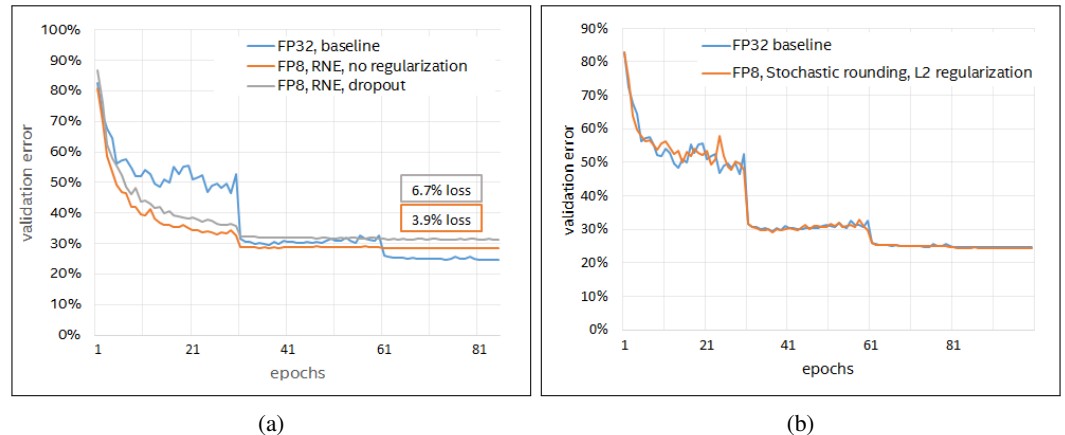

(a)                                         (b)

Figure 4: (a) Comparing validation performance with 'dropout' and noise-based implicit regularization techniques using RNE(round to nearest even) (b) model performance with stochastic rounding with L2 regularization.

in the forward, backward, weight update paths for all convolution and GEMM kernels as described in Section.3.

Using the proposed training method, we have successfully trained two residual networks (Resnet-34/50) (He et al., 2016) on Imagenet-1K (Deng et al., 2009) data set. For 8-bit training experiments, we have quantized the Convolution, BatchNorm (stats are accumulated at full precision), ReLU and EltWise layers, while the data layer and SoftMax layers use full precision. We have maintained the first Convolution layer, the following BatchNorm layer, and final Dense layer at half-precision to minimize the accuracy loss. The first convolution layer is expected to have a minimum of eight mantissa bits to accurately represent the normalized image data passed from the data layer. We also store the master copy of weights in half-precision; this can potentially speed up weight update operation by up to $2\times$, because the operation is memory bound. Since the weight update computation is performed at full precision (as descried in Section.3), to realize these bandwidth benefits the up/down conversion of the master weights between full and half precision formats must be strictly performed in the cache or near memory. We have used identical batch size (256), and hyper parameters for both full precision and 8-bit training experiments. Both sets of experiments also employ 'warm up' for the first 5 epochs, with an initial learning rate of 0.128, and train for a total of 90 epochs to achieve the final accuracy.Table.2 summarizes the validation accuracy achieved by convolution networks on imagenet-1K (Deng et al., 2009) dataset.

Table 2: Top-1 validation accuracy for convolution networks on Imagenet-1KDeng et al. (2009).

| Model | Dataset | Batch-size | Epochs | FP32 (top-1 %) | FP8 (top-1 %) |
|---|---|---|---|---|---|
| Resnet-50 | imagenet-1K | 256 | 90 | 76.38 | 76.18 |
| Resnet-34 | imagenet-1K | 256 | 90 | 73.47 | 73.18 |
| Resnet-18 | imagenet-1K | 256 | 90 | 69.23 | 69.71 |

Figure.5 shows the convergence plots for Resnet-34 and Resnet-50 comparing top-1 accuracy of FP8 training with the baseline FP32 training. It can be seen that the validation accuracy of FP8 training closely follow the baseline numbers indicating the robustness of the training method.

In addition to convolution networks, we have also trained two state of the art machine translation workloads (GNMT (Wu et al., 2016) and Transformer (Vaswani et al., 2017)) and demonstrated BLEU scores matching single precision baselines. We trained an 8-layer GNMT (Wu et al., 2016) encoder/decoder LSTM model with 1024 recurrent units and 1024 attention units. We trained this network using 8-bit floating point format for all MatMul operations, including the LSTM, attention modules, embedding lookup is performed at 8-bit while the updates are computed at full precision. The activation functions such as tanh and sigmoid use half-precision data type. Our 8-bit training

Table 3: Comparison of our method with the only other FP8 training method on Imagenet-1K (Deng et al., 2009) data set. W, A, E, G, MasterWts represent the precision setting for weights, activations, error, weight gradients and mater copy of weights respectively.

| Method, Format | W,A,E,G | MasterWts | Resnet-18 (top-1 error %) | Resnet-50 (top-1 error %) |
|---|---|---|---|---|
| Wang et al. (2018), FP8 | 8,8,8,8 | 16 | 33.05 | 28.28 |
| Ours, FP8 | 8,8,8,8 | 16 | 30.29 | 23.82 |

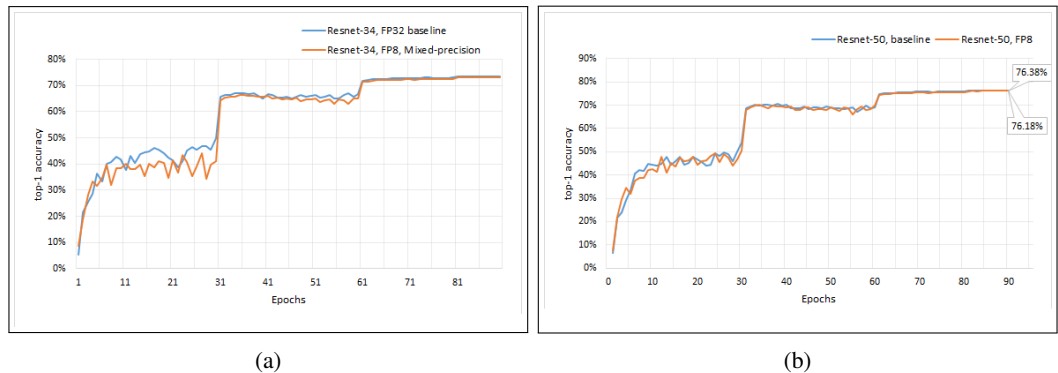

(a)            (b)

Figure 5: Convergence plots showing Top-1 validation accuracy for. (a) Resnet-34 (He et al., 2016) (b) Resnet-50 (He et al., 2016) on imagenet-1K (Deng et al., 2009) dataset.

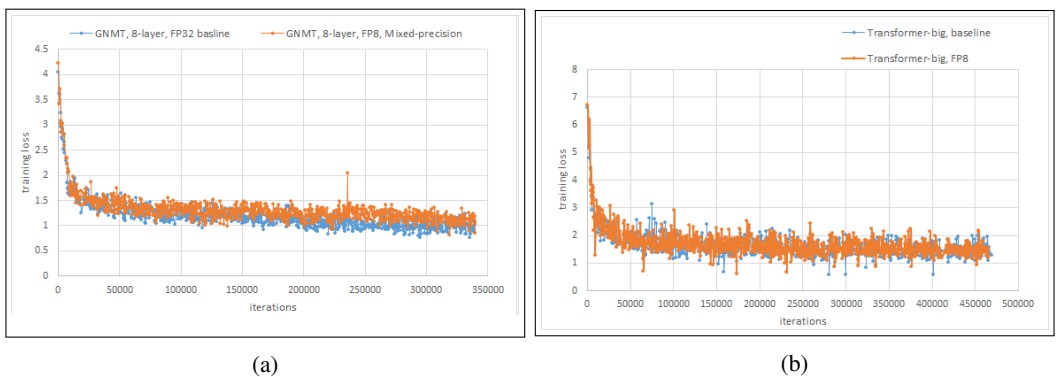

(a)            (b)

Figure 6: Convergence plots showing training loss for (a) 8-layer GNMT (Wu et al., 2016) and, (b) 6-layer Transformer (Vaswani et al., 2017) trained on WMT16 English → German dataset.

experiments use identical batch size and hyper parameters to that of the full precision baseline and the network is trained for 10 epochs. We used the loss scaling schedule described in Section.3.1.

We also trained a 6-layer Transformer-big (Vaswani et al., 2017) translation network with roughly 200M parameters. For the 8-bit experiments, we quantized all the MatMul operations in the encoder, decoder, including attention layers. Embedding lookup operation is performed at 8-bit precision while the updates are computed at full precision. We use the identical configuration for both baseline and 8-bit training experiments. Both GNMT (Wu et al., 2016) and Transformer (Vaswani et al., 2017) models were trained on large scale, WMT2016 English→German dataset consisting of 4.5 million sentence pairs. The results are summarized in Table.4.

## 5 CONCLUSION

We demonstrate state-of-the-art accuracy across multiple data sets (imagenet-1K, WMT16) and a broader set of workloads (Resnet-18/34/50, GNMT, Transformer) than previously reported. We

Table 4: sacreBLEU (Post, 2018) score measured on WMT 2014 English→German dataset

| Model | Dataset/ Task | Epochs | FP32 baseline | FP8 Mixed Precision |
|---|---|---|---|---|
| GNMT | WMT 2016 English→German | 10 | 24.3 | 24.6 |
| Transformer-big | WMT 2016 English→German | 10 | 27.59 | 27.33 |

propose easy to implement and scalable solution for building FP8 compute primitives, eliminating the need for stochastic rounding hardware in the critical compute path, as proposed by Wang et al. (2018), thereby reducing the cost and complexity of the MAC unit. We explore issues around gradient underflow and quantization noise that arise as a result of using the proposed 8-bit numeric format for large scale neural network training. We propose solutions to deal with these problems in the form of enhanced loss scaling and stochastic rounding.

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
