# OpenReview forum: "Mixed Precision Training With 8-bit Floating Point"
_ICLR.cc/2020/Conference — Reject_

### Official Review · AnonReviewer4 · 2019-10-18
**Official Blind Review #4**

**Rating:** 1

**Review:**

This paper is about training deep models with 8-bit floating point numbers. The authors use an enhanced loss scaling method and stochastic rounding method to stabilize training. They do experiments on image classification and NLP tasks.

The paper is clearly written. However, I don’t think this paper passes the bar of ICLR. This paper lacks innovation and insightful analysis.

1.Sec. 3.1 proposes enhanced loss scaling. Loss scaling is a heuristic to train low-precision neural networks. The authors train 8-bit GNMT with a changing scaling factor. However, this looks like some manually tuned result for GNMT only. I doubt if this generalizes to other models. Besides, there is no equation or algorithm flowchart to demonstrate their method. It’s not very readable.

2.The logic of Sec. 3.2 is quite confusing. The authors first empirically show that the performance of ResNet-50 significantly drops with 8-bit training. Then they show the sum of the square of the weights in ResNet-50 is high at the beginning. With this observation, they claim it demonstrates the drawback of ‘rounding-to-nearest-even’. I cannot see the connection between the norm of weights and the rounding technique. Moreover, the stochastic rounding has already been used in 8-bit training.[1]

3.The setting in the experiment section is not stated clearly. For example, what’s the hyper-parameter for loss scaling? Another question is the gradient. In Sec. 3, just above Fig. 1, the authors claim the weight update is performed in full-precision. In contrast, they claim the gradient is 8-bit in table 3. If the update is full-precision, [2] is an important baseline.

Small suggestions:
1.For Fig. 6, I suggest the authors to smooth the loss curves to avoid overlap of two curves.
2.There are two ‘with’s in the last paragraph of page 7.

Reference:
[1]Wang N, Choi J, Brand D, et al. Training deep neural networks with 8-bit floating point numbers[C]//Advances in neural information processing systems. 2018: 7675-7684.
[2]Banner R, Hubara I, Hoffer E, et al. Scalable methods for 8-bit training of neural networks[C]//Advances in Neural Information Processing Systems. 2018: 5145-5153.

**Experience Assessment:**

I have read many papers in this area.

**Review Assessment: Checking Correctness Of Derivations And Theory:**

I carefully checked the derivations and theory.

**Review Assessment: Checking Correctness Of Experiments:**

I carefully checked the experiments.

**Review Assessment: Thoroughness In Paper Reading:**

I read the paper at least twice and used my best judgement in assessing the paper.

---

> ### Author Response · Authors · 2019-11-14
> **Response to AnonReviewer4**
>
> Thank you for your detailed review and comments.
> Q1.
> Please note that only GNMT required the hand tuned loss scaling schedule. We believe this method can be automated for GNMT as well. We have observed that GNMT saw wider error gradient distributions which often consisted of outliers that are much larger than the mean. When these outliers are scaled with a large scaling factor, they overflow and cause a NaN when gradients for previous layer are computed. The current automatic loss scaling algorithm is ill-equipped to handle these transient NaNs, it over-corrects (reduces) the loss scale value every time it encounters an outlier, resulting in divergence. Our enhanced loss scaling strategy mitigates this by adding a lower threshold to prevent loss scale value from becoming too small. We believe adding a few additional conditions to loss scaling algorithm will handle this case automatically.
> The current loss scaling algorithm works like this:
> Initial ‘loss_scale’ value is set to ‘max_threshold’.
> When a gradient computation results in a NaN, reduce the loss_scale by a factor of ‘scale’ (=2)
> If there is another NaN within the ‘interval’, the loss scale is further reduced by a factor of 2.
> If there is no NaN encountered for ‘interval’ (=2000) iterations, the ‘loss_scale’ value is increased by a factor of 2
> When the gradients have lot of outliers, we would see more of these spurious NaNs and the ‘loss_scale’ value quickly drops. One or more of the following solutions can be applied to solve this.
> 1.	Reduce ‘interval’ to a smaller iteration count (=200) so the ‘loss_scale’ value can recover to quickly from a previous drop.
> 2.	Ignore a few NaNs unless they appear in consecutive iterations. This will address the over-correction  (similar to setting a lower threshold)
> 3.	A more generic solution is to derive layer-wise scaling factor which is aware of the gradient distribution at each layer [1]
> As per your feedback, we will update the paper with a description and/or a flow chart of this algorithm.
>
> Q2. On connection between the norm and rounding technique.
>
> As we discussed in Section 3.2, rounding plays a significant role in FP8 training because rounding errors are quite large at this precision. It is known that round to nearest even (RNE) distorts the data distribution to have more even numbers than odd. As a result of this when using RNE, rounding errors grow at the rate proportional to square root of number of accumulations. (more here: https://en.wikipedia.org/wiki/Rounding#Floating-point_rounding)
> In Figure 3c, we are showing the result of these accumulated errors on the weight distribution. The overall weight distribution is shifted towards larger numbers resulting in increasing “L2_loss” (=sum of squares of the weights). Since l2_loss is used as a ‘regularization’ term ( loss =cross_entropy+l2_loss), the loss increases as the rounding errors keep accumulating. This leads to loss of generalization, as shown in Figure 3a and 3b – the training loss keeps going down while validation loss is increasing.
>
> To avoid using l2_loss term, we tried using ‘drop out’ method and trained without any regularization. Though the validation error improved in both these cases, there was still a significant gap in final accuracy due to ineffectiveness of these regularization methods.
>
> Then then we went back to l2 regularization – this time addressing the rounding errors in the gradients using stochastic rounding. This helped keep the accumulation of errors in check and the we achieved SOTA accuracy.
>
> Q3.
> The single hyper-parameter used for loss scaling indicates whether to use a ‘static’ or a ‘dynamic’ loss scaling method. We will add this detail to experiments section.
>
>  Q3b. On the relevance of Banner et.al. [3] as an important baseline.
>
> In our case the update is not full precision. We compute weight gradients at FP8 precision and we use FP8 weight gradients and FP16 master weights for the weight update operation. In Figure 1 we are showing FP32 because the internal accumulator in ALU unit is FP32, during weight update the weights are accumulated into FP32 accumulator and are converted to FP16 before they are written out to the master copy, we have described this in Section 3, para 3.
> In contrast Banner et.al [3] use a technique called ‘gradient bifurcation’ where they only quantize one of the two convolutions in the backward pass. They maintain two copies of the error gradient one of which is at full precision. The full precision copy is used to compute the error gradients at FP32 precision and passed down to the previous layer.
> Hope that helps clarify your questions.
>
> [1] Adaptive Loss Scaling for Mixed Precision Training, Ruizhe Zhao, Brian Vogel, Tanvir Ahmed
> [2] Wang N, Choi J, Brand D, et al. Training deep neural networks with 8-bit floating point numbers
> [3] Banner R, Hubara I, Hoffer E, et al. Scalable methods for 8-bit training of neural networks,

---

> > ### Comment · AnonReviewer4 · 2019-11-14
> > **Thank You For Your Response**
> >
> > A1. If you have an algorithm for your method, I would like to see it in your paper (e.g., write a pseudocode) instead of in the rebuttal.
> > Technically, your method is able to address overflow. However, by dividing the loss factor by 2, the method will cause underflow. What if overflow and underflow happen at the same time? This is an important issue to address. If this is impossible for training GNMT, could you show some evidence? Moreover, GNMT is king of old that I doubt the value of a method that only works for GNMT.
> >
> > A2. Please distinguish your stochastic rounding method with reference [1]. I cannot see the novelty of this part.
> >
> > Overall, after reading your response, it seems that Sec 3.1 is only for GNMT and Sec 3.2 is only for ResNet 50. The motivation now looks confusing. The novelty of this paper is still not clarified. My suggestion is that you should polish your paper and include more insights and novelty.

---

> > > ### Author Response · Authors · 2019-11-15
> > > **Thank you for your quick feedback**
> > >
> > > >> What if overflow and underflow happen at the same time? This is an important issue to address
> > >
> > > Yes, this can happen. It happens and more frequently with GNMT in the early epochs. This is the exact issue we are addressing with enhanced loss scaling.
> > >
> > > We see more frequent gradient overflows because of a few outliers in the distribution, while a significant chunk of the gradients experience underflow.
> > > This happens more frequently with GNMT because it does not use any normalization layers which lead to more irregular data distributions. Also, RNNs tend to accumulate errors quickly compared to feed-forward networks, this is exacerbated by the additional noise induced by the low precision (FP8).
> > >
> > > The existing loss scaling algorithms treat all overflows equally -- if they see an overflow, they drop the scaling factor. This leads to scaling factor dropping very quickly because of these spurious outliers.
> > > The fix we proposed to our algorithm is to ignore a few spurious overflows which are likely a result of the outliers and continue to maintain a higher loss scale value. We accomplish this by setting a ‘lower threshold’ for the loss scale value to prevent it from going below a certain threshold value even when overflows occur – and this strategy worked as evidenced by the GNMT result.
> > >
> > > Now, to automate this process, we will add a new variable ‘consecutive_overflow_threshold’, (=2 or 3 depending on the workload). This will enable the loss scaling algorithm to ignore overflows unless they occur in succession for ‘consecutive_overflow_threshold’ times, which is a more reliable indicator of a true shift in the gradient distribution, and not caused by spurious outliers. We will also reduce the interval between loss scale updates (from 2000 to 500), so there is a better chance to recover from any inadvertent drop in loss scale value.
> > >
> > > >> Moreover, GNMT is kind of old that I doubt the value of a method that only works for GNMT.
> > >
> > > GNMT is kind of old, but It is also more difficult to converge at low precision because of the reasons discussed above. This is not the case for feed forward networks that include layer normalization as evidenced by our Transformer result.  Based our observations, we believe automatic loss scaling will work for a large percentage of the feed-forward networks.
> > >
> > > We have updated the paper with the pseudo code for enhanced loss scaling algorithm.
> > >
> > > >> it seems that Sec 3.1 is only for GNMT and Sec 3.2 is only for ResNet 50. The motivation now looks confusing.
> > > Section 3.1 is mostly addressing loss scaling issues of GNMT because other networks we converged did not have any issues with existing loss scaling method. We think GNMT represents kind of an extreme case for the following reasons:
> > > 1.	it is a recurrent network which tend accumulate gradient errors quickly, which is exacerbated by the noise induced by low-precision.
> > > 2.	It does not use any kind of normalization layers, leading to more irregular data distributions, which are difficult handle for the standard loss scaling algorithms.
> > >
> > > The observations from Section 3.2  are applicable across the workloads – we have chosen Resnet-50 as an example to clearly demonstrate the effects of noise on generalization and how that can be addressed with stochastic rounding.  We have observed similar behavior across all three workloads we have demonstrated – and they all use stochastic rounding for the gradients. We have not added additional plots for Transformer and GNMT in the interest of space.
> > >
> > > >> Please distinguish your stochastic rounding method with reference [1].
> > >
> > > Stochastic rounding is not new, the difference is in how it was implemented.  Our implementation is more efficient for the following reasons:
> > > 1.	We perform stochastic rounding only ‘once’ after the full MatMul operation is complete. Wang et.al perform stochastic rounding on the accumulator after every few (8 to 32) FMA instructions. This incurs a few orders of magnitude higher overhead compared to our implementation depending on the number of FMA instructions required by MatMul . They also need to replicate this capability inside each FMA unit which costs more power and silicon area.
> > > 2.	 Our rounding method itself is more efficient because we use 8-bit PRNG (LFSR) for generating the random probability. We also reuse the random numbers quite extensively ( > 256 times). This reduces the cost of stochastic rounding hardware quite significantly.
> > > We contribute the following to the state-of-the art FP8 training.
> > > -	We show better coverage across multiple datasets & workloads. As a result, we uncovered issues like gradient noise and loss scaling and propose solutions to handle them.
> > > -	We proposed a better and more efficient approach to implementing  FP8 hardware compared to the one proposed by Wang et.al.
> > > - Previous results from Wang et.al. only show results for Resnet 50.
> > > Hence we believe there is significant novelty in the work we presented. Hope that addresses your questions.

---

> > > > ### Comment · AnonReviewer4 · 2019-11-15
> > > > **Still need improvement to be publishable in ICLR**
> > > >
> > > > Thank you for your quick response.
> > > >
> > > > I think the method part of the paper still needs much improvement to clarify the novelty and contribution. Also, the experiments in the paper are not enough to demonstrate the generalizability of the proposed methods across different models and datasets. In its current form, I'm afraid the paper cannot reach the borderline as an ICLR paper. However, I think the paper do has potential. You can submit it to other conferences after polishing the method part and adding more supportive experiments.

---

### Official Review · AnonReviewer3 · 2019-10-21
**Official Blind Review #3**

**Rating:** 6

**Review:**

In this paper, the authors propose a method to train deep neural networks using 8-bit floating point representation for weights, activations, errors, and gradients. They use enhanced loss scale, quantization and stochastic rounding techniques to balance the numerical accuracy and computational efficiency. Finally, they get a slightly better validation accuracy compared to full precision baseline. Overall, this paper focuses on engineering techniques about mixed precision training with 8-bit floating point, and state-of-the-art accuracy across multiple data sets shows the effectiveness of their work.

However, there are some problems to be clarified.
1. The authors apply several techniques to improve the precision for training with 8-bit floating point, but they do not show the gain for each individual. For example, how much improvement can this work achieve when just using enhanced loss scaling method or a stochastic rounding technique? This should be clearly presented and more experimental comparison is expected.

2. The paper should present a bit more background knowledge and discussion on the adopted techniques. For instance, why the stochastic rounding method proposed in this article by adding a random value in probability can regulate quantization noise in the gradients? And why Resnet-50 demands a large scaling factor?

3. On Table 3, in comparison with Wang et al. (2018), the authors use layers with FP32 (not FP16 in Wang). Thus, it is hard to say the improvement comes from the proposed 8-bit training. This should be clarified.

4. How to set the hyper-parameters, such as scale, thresholds and so on, is not clear in the paper. There are no guidelines for readers to use these techniques.

5. The authors did not give a clear description of the implement for the enhanced loss scaling. They apply different loss scaling methods for different networks. This should be explained in detail.

6. In the experiment, for a single model, some layers are 8-bit, some layers are 32-bit and some layers are 16-bit.  Is the 8-bit training only applicable for a part of the model?  How do we know which layer is suitable for 8-bit training?

**Experience Assessment:**

I have read many papers in this area.

**Review Assessment: Checking Correctness Of Derivations And Theory:**

N/A

**Review Assessment: Checking Correctness Of Experiments:**

I assessed the sensibility of the experiments.

**Review Assessment: Thoroughness In Paper Reading:**

I read the paper at least twice and used my best judgement in assessing the paper.

---

> ### Author Response · Authors · 2019-11-14
> **Response to AnonReviewer3**
>
> Thank you for your comments. We will attempt to answer your questions below.
> Q1.
> Our intention was to show that both enhanced loss scaling and stochastic rounding are essential for achieving full accuracy with FP8 training.
> For example, in section 3.1, our experiments already use “stochastic rounding” on gradients (essential for convergence) to study the impact of loss scaling in isolation.
> Similarly, in section 3.2 when studying the impact of stochastic rounding, we employed the ‘best loss scaling strategy’ derived from section 3.1. Perhaps this is not clearly described in the paper. We will edit the text for clarity and upload the new version of the paper.  In Figure 2a, we have compared multiple experimental results to demonstrated impact of using different loss scaling values on final accuracy of Resnet50.
>
> Q2.
> On stochastic rounding : As we discussed in Section 3.2, rounding plays a significant role for FP8 because the rounding errors are quite large at this precision. It is known that standard rounding methods (up, down, towards zero, away from zero) have a positive or a negative bias to the final distribution. The most popular rounding method used by floating point today is round to nearest even (RNE) – although this method is free of positive or a negative bias -- it distorts the data distribution to have more even numbers than odd. (more info here: https://en.wikipedia.org/wiki/Rounding#Floating-point_rounding). It is also known that rounding errors grow with longer accumulation chains (like in Convolution and MatMul). For RNE method, the rounding errors grow proportional to the square root of number of accumulations. This is quite significant at extreme low precisions (like FP8) where ‘episilon’ value is large.
>
> Stochastic rounding is bias free because it uses random probability term for tie-breaking. It does not impact the overall data distribution of the tensor and the rounding errors are small and evenly distributed. This makes the accumulation of errors during long accumulation chains much less likely.
> >> On why Resnet-50 demands a large scaling factor? :
> In general working with FP8 would require larger scaling factor because FP8 has smaller dynamic range compared to FP16. The smallest number that can be represented by FP16 is 5.96e-8 whereas the smallest number that FP8 can represent is 1.52e-5. This means that a larger percentage of smaller gradients fall ‘below’ the FP8 range. Hence, we need to use a larger scaling factor to push them up into the FP8 range.
>
> Q3.
> We would like to clarify that we do not use FP32 in any of our training results. For Resnet-50 , all convolution and batchnorm layers use FP8 -- except the first conv and last FC layers which use FP16; we also use FP16 master copy of weight. This configuration identical to what is used by Wang et.al.-- hence the comparison is fair.
> The key difference between our implementations is that we use FP32 accumulator (in the ALU) while Wang et.al use a modified FP16 (1-6-9 format) – as a result, they need to implement additional hardware in the ALU path to perform stochastic rounding on the accumulator to preserve accuracy. Given the complexity of building stochastic rounding hardware, their implementation will be more expensive to build. We discussed these design trade-offs in Section 1.
>
> Q4.
> We employ the widely disseminated techniques that are used for FP16 mixed precision training, these are implemented in frameworks such as Tensorflow and PyTorch. Our loss scaling methods are modifications on top of these baseline methods.To answer your specific question : Scale (=2) and threshold (min=2, max=2^14) values are hard-coded in in the current implementation of loss scaling algorithm. The dynamic loss scaling algorithm increments the loss scale value by a factor of ‘scale’ every 2000 iteration intervals and reduced the loss scale by a factor ‘scale’ in the case of an occurrence of ‘NaN’ in the during gradient computation.  For GNMT training, the enhanced loss scaling method updates the ‘min’ threshold value according to the schedule shown in Figure 2b to prevent the loss scale becoming too small. We will add the description of the algorithm to the paper.
>
> Q5.
> We have described the loss scaling methods applied to each model in section 3.1
> For Resnet50, we use constant loss scaling of 10K, this is derived empirically through experimentation which are detailed in section 3.1. For GNMT and Transformer, we use dynamic loss scaling implemented by Tensorflow.
>
> Q6.
> For now, the process of selecting which layers to run at FP8 requires human expertise and intervention. But we expect the future frameworks to automate this process of selecting multiple precision options to maximize performance. Recent work on use of AutoML [1] for mixed-precision quantization is also promising research direction.
> [1] HAQ: Hardware-Aware Automated Quantization with Mixed Precision, Kuan Wang et.al., CVPR 2019.

---

### Official Review · AnonReviewer1 · 2019-11-02
**Official Blind Review #1**

**Rating:** 6

**Review:**

Originality: The paper proposed a new scaling loss strategy for mixed-precision (8-bit mainly) training and verified the importance of rounding (quantization) error issue for low-precision training.

Quality: The authors clearly illustrated the benefit of their proposed loss strategy and the importance of quantization error for two different tasks (image classification and NMT). The experiments are very clear and easy to follow.

Clarity: The paper is clearly written with some visualizations for readers to understand the 8-bit training.

Significance:
1. The enhanced loss scaling strategy is interesting but the method seems hand-tuning. Is there any automatical way or heuristic deciding way?
2. The stochastic rounding method is very intuitive. How do you choose the value of "r" in the equation? Is it a sensitive hyper-parameter or not?

Typos:
Page 7: with with roughly 200M ->  with roughly 200M


**Experience Assessment:**

I do not know much about this area.

**Review Assessment: Checking Correctness Of Derivations And Theory:**

N/A

**Review Assessment: Checking Correctness Of Experiments:**

I carefully checked the experiments.

**Review Assessment: Thoroughness In Paper Reading:**

I read the paper thoroughly.

---

> ### Author Response · Authors · 2019-11-14
> **Response to AnonReviewer1**
>
> Thank you for your helpful comments.
> Q1: The enhanced loss scaling strategy is interesting but the method seems hand-tuning. Is there any automatical way or heuristic deciding way?
>
> We believe this can be automated. We have observed that GNMT saw wider error gradient distributions which often consisted of outliers that are much larger than the mean. This is exacerbated by the additional noise induced as a result of using lower precision (FP8) for error gradients. When these outliers are scaled with a large scaling factor, they overflow and cause a NaN when gradients for previous layer are computed. The current automatic loss scaling algorithm is ill-equipped to handle these transient NaNs, it over-corrects (reduces) the loss scale value every time it encounters an outlier, resulting in divergence. Our enhanced loss scaling strategy mitigates this by adding a 'minimum threshold' to prevent loss scale value from becoming too small. We believe adding a few additional conditions to loss scaling algorithm will handle this case automatically.
>
> The current loss scaling algorithm works like this:
> Initial ‘loss_scale’ value is set to ‘max_threshold’.
> When a gradient computation results in a NaN, reduce the loss_scale by a factor of ‘scale’ (=2)
> If there is another NaN within the ‘interval’, the loss scale is further reduced by a factor of 2.
> If there is no NaN encountered for ‘interval’ (=2000) iterations, the ‘loss_scale’ value is increased by a factor of 2
>
> When the gradients have lot of outliers, we would see more of these spurious NaNs and the ‘loss_scale’ value quickly drops. One or more of the following enhancements can be applied to automatic loss scaling algorithm to address this:
>
> 1.	Reduce ‘interval’ to a smaller iteration count (=200) so the ‘loss_scale’ value can recover to quickly from a previous drop.
> 2.	Ignore a few NaNs unless they appear in consecutive iterations. This will address the over-correction  (similar to setting a lower threshold)
> 3.	A more generic solution is to derive layer-wise scaling factor which is aware of the gradient distribution at each layer [1]
>
> Q2: The stochastic rounding method is very intuitive. How do you choose the value of "r" in the equation? Is it a sensitive hyper-parameter or not?
>
> We appreciate the positive feedback.
> The value of “r” is an 8-bit random number generated using LFSR random number generator. We also reuse these random numbers (for about 256 times) to save on the overheads to generate these numbers.
>
> We will fix the typos and grammatical errors you pointed out and update the paper.
>
> Hope this clarifies your questions.
>
> [1] Adaptive Loss Scaling for Mixed Precision Training, Ruizhe Zhao, Brian Vogel, Tanvir Ahmed (https://arxiv.org/pdf/1910.12385)

---

### Author Response · Authors · 2019-11-15
**Updated revision of the paper**

We have added an updated version of the paper with the following changes:

> Description and pseudo code for enhanced loss scaling algorithm (section 3.1)
> Fixed typographical and grammatical errors pointed out by the reviewers (section 4, page 8)

We thank all the reviewers for their helpful comments and feedback.

---

### Decision · Program_Chairs · 2019-12-19

**Decision:**

Reject

**Comment:**

This paper propose a method to train DNNs using 8-bit floating point numbers, by using an enhanced loss scaling method and stochastic rounding method. However, the proposed method lacks novel and both the paper presentation and experiments need to be improved throughout.